# The Alternative Splicing Landscape of *Brassica napus* Infected with *Leptosphaeria maculans*

**DOI:** 10.3390/genes10040296

**Published:** 2019-04-11

**Authors:** Jin-Qi Ma, Li-Juan Wei, Ai Lin, Chao Zhang, Wei Sun, Bo Yang, Kun Lu, Jia-Na Li

**Affiliations:** 1Chongqing Rapeseed Engineering Research Center, College of Agronomy and Biotechnology, Southwest University, Chongqing 400715, China; jinqima1996@163.com (J.-Q.M.); lijuan525888@163.com (L.-J.W.); ai_linnn@163.com (A.L.); 18083606406@163.com (C.Z.); reginasw@163.com (W.S.); sheepneck@hotmail.com (B.Y.); drlukun@swu.edu.cn (K.L.); 2Academy of Agricultural Sciences, Southwest University, Chongqing 400715, China

**Keywords:** alternative splicing, RNA-seq, WGCNA, plant defense, transcription factors

## Abstract

Alternative splicing (AS) is a post-transcriptional regulatory process that enhances transcriptome diversity, thereby affecting plant growth, development, and stress responses. To identify the new transcripts and changes in the isoform-level AS landscape of rapeseed (*Brassica napus*) infected with the fungal pathogen *Leptosphaeria maculans*, we compared eight RNA-seq libraries prepared from mock-inoculated and inoculated *B. napus* cotyledons and stems. The AS events that occurred in stems were almost the same as those in cotyledons, with intron retention representing the most common AS pattern. We identified 1892 differentially spliced genes between inoculated and uninoculated plants. We performed a weighted gene co-expression network analysis (WGCNA) to identify eight co-expression modules and their Hub genes, which are the genes most connected with other genes within each module. There are nine Hub genes, encoding nine transcription factors, which represent key regulators of each module, including members of the *NAC*, *WRKY*, *TRAF*, *AP2/ERF-ERF*, *C2H2*, *C2C2-GATA*, *HMG*, *bHLH*, and *C2C2-CO*-like families. Finally, 52 and 117 alternatively spliced genes in cotyledons and stems were also differentially expressed between mock-infected and infected materials, such as *HMG* and *C2C2-Dof*; which have dual regulatory mechanisms in response to *L. maculans*. The splicing of the candidate genes identified in this study could be exploited to improve resistance to *L. maculans*.

## 1. Introduction

Alternative splicing (AS) was first discovered by Gilbert in 1978 [1]. During this post-transcriptional regulatory process, precursor-mRNA (pre-mRNA) produces differentially spliced RNA transcripts that may be translated into diverse protein isoforms [2]. AS events are quite common in many organisms, occurring in 92–94% of intron-containing genes in human (*Homo sapiens*) [3], 60% in *Arabidopsis thaliana* [4], 52% in soybean (*Glycine max*) [5], 40% in *Gossypium raimondii* [6], 40% in maize (*Zea mays*) [7], and 33% in rice (*Oryza sativa*) [8]. The various types of AS events include the formation of skipped exons, retained introns, alternative 5′ splice sites, alternative 3′ splice sites, mutually exclusive 3′ UTRs, tandem UTRs, mutually exclusive 5′ exons, and mutually exclusive exons, all of which increase the complexity of the transcriptome and the diversity of the proteome. AS is prevalent in eukaryotic organisms [9], but the types of AS events vary among species. For example, skipped exons are the most common type of AS event, and retained introns are the least common in animals and yeast [10,11], whereas retained introns are the most common type of AS event in plants [5]. AS also exhibits tissue-specific patterns [12]. In maize, these patterns are related to the sensitivity of genes to miRNAs, methylation [13], and nonsense-mediated decay [14]. For example, isoforms of the SPX (SPX domain-containing protein) gene that are sensitive to miR827 are common in developing seedlings [7]. The production of premature termination codons by AS and the subsequent degradation of mRNA via nonsense-mediated decay or miRNAs is thought to represent a mechanism for fine-tuning gene expression [15].

The regulation of AS is based on spliceosomes, which are formed by small nuclear ribonucleoprotein (snRNP) modules (U1, U2, U4, U5, and U6 snRNPs), cis-acting sequences, and trans-acting RNA binding proteins (RBPs) [16]. The cis-acting sequences consist of two motifs: splicing signals (SSs) and splicing regulatory elements (SREs). The SS is strictly required for consistent splicing, including at the 5′ splice site, the branch point, and the 3′ splice site. Nearly all plant introns possess a 5′ GU, a 3′ AG, and an A at the branch point, a characteristic known as the “GU—AG” rule [17]. RBPs recognize multiple weak, degenerate signals in SREs, including exonic splicing enhancers, exonic splicing silencers, intronic splicing enhancers, and intronic splicing silencers [18]. The process of consistent splicing involves the interaction of RBPs and SSs: U1 recognizes the 5′ splice site, U2 recognizes the branch point, and U2AF proteins recruit U2 to recognize, splice, and assemble the polypyrimidine tract near the 3′ splice site [19]. The regulatory mechanism of AS has been widely studied in human, yeast (*Saccharomyces cerevisiae*), and mouse (*Mus musculus*), in which the interaction between RBPs and SREs is important for facilitating or inhibiting the assembly of the spliceosome [20] via processes termed exon definition and intron definition.

The diverse mRNA and protein isoforms produced by AS lead to changes in their structure, function, localization, and other biochemical properties [21]. AS functions in various processes [22] in plant development, such as cell fate determination, flowering, and the circadian clock, as well as in plant responses to biotic/abiotic stresses, such as heat, drought, and disease [23,24,25]. For example, it was reported that constitutive and inducible alternative splicing of *OsWRKY62* and *OsWRKY76* may defend the blast fungus *Magnaporthe oryzae* and the leaf blight bacterium *Xanthomonas oryzae* pv *oryzae* [26].

Understanding the interaction between plants and pathogens is critical for breeding disease-resistant plants. AS is a crucial mechanism used by plants to detect pathogen attack and trigger plant immunity [27]. *Leptosphaeria maculans* [28], a hemibiotrophic fungal pathogen, is the causal agent of the stem canker disease known as blackleg in *Brassica napus*, the second most widely produced oilseed crop worldwide [29]. *B. napus*, known as oilseed rape, is an allopolyploid species through spontaneous hybridizations between *Brassica rapa* and *Brassica oleracea* [29]. There are 101,040, 41,174 and 45,758 annotated genes of *B. napus*, *B. rapa* [30] and *B. oleracea* [31]. *L. maculans* of *B. napus* causes serious economic losses in Europe, Australia, North America, and China. It was estimated that UK losses of £56M per season reported in 2006 [28] and an average losses of £235M each year (http://www.cropmonitor.co.uk/) for the period 2005–2014 caused by phoma canker (*L. maculans* and *L. biglobosa*). Many genes associated with disease resistance produce AS isoforms that recognize the invasion of diverse pathogens and launch immune responses [32]. For example, AS of the translation initiation factor 4E in tomato (*Solanum lycopersicum*) functions in plant defense again Potato virus Y and Pepper mottle virus [33]. The AS isoforms of SCL33 in *Brachypodium distachyon*, a serine/arginine-rich splicing factor, function in the response to *Panicum mosaic virus* and its satellite virus [2]. However, the characteristics and functions of AS between *B. napus* and *L. maculans* remain unknown.

Current studies have exploited the molecular mechanism, the function and research method of AS in many plants, especially in the model plant *A. thaliana*, which is an ancestral karyotype to *B. napus*. In this study, we present an isoform-level AS landscape of *B. napus* infected with *L. maculans* to extend AS research and shed light on the response of *B. napus* to this agronomically important pathogen.

## 2. Materials and Methods

### 2.1. Downloading of RNA-seq Data

The RNA-seq data were downloaded from NCBI using Aspera [34]. All raw data were downloaded in NCBI GEO with the accession number GSM2175146, GSM2175147, GSM2175148, GSM2175149, GSM2175150, GSM2175151, GSM2175152 and GSM2175153. The data were obtained from various tissues, including cotyledons and stems of *B. napus* cultivar Darmor-bzh mock-inoculated and inoculated with *L. maculans* in two biological replicates: GSM2175146 and GSM2175147 (non-inoculated cotyledons); GSM2175148 and GSM2175149 (inoculated cotyledons); GSM2175150 and GSM2175151 (non-inoculated stems); and GSM2175152 and GSM2175153 (inoculated stems).

### 2.2. Assembly of Putative Transcripts and Analysis of the AS Landscape

After converting the raw data with the SRA Toolkit 2.9.0 (https://www.ncbi.nlm.nih.gov/Traces/sra/?view=software) [35] and performing quality trimming with Trimmomatic-0.36 [36] (modified parameters: ILLUMINACLIP:TruSeq3-PE.fa:2:30:10, LEADING:3, TRAILING:3, and SLIDINGWINDOW:4:15), the prefiltered reads were mapped to the *B. napus* reference genome with STAR-2.5.3a (with sjdbOverhang = 150 and limitBAMsortRAM = 6000000000) [37]. Transcript assembly was performed using Cufflinks-2.2.1 [38]. Using Cuffcompare, the different isoforms were mapped to the corresponding genes. The Astalavista-4.0 [39] tool was used to analyze the AS patterns and to visualize the AS landscape. To identify the homologous genes in *A. thaliana* of the differentially spliced genes in *B. napus*, we performed blast analysis using TAIR database (https://www.arabidopsis.org/index.jsp).

### 2.3. Weighted Gene Co-expression Network Analysis of Overlapping Differentially Spliced Genes

To investigate the functions of overlapping differentially spliced genes, WGCNA [40] in R version 3.4.4 was performed, including sample clustering, outlier detection, soft threshold filtering, one-step network construction, module identification, and module relationship analysis. From the above analysis, we gained the weighted values, which represent the relationships between genes in pairs. The weighted values, which are >0.15, were used to perform network analysis and gain the genes “degree (i.e., the number of genes linked to the gene)” using Cytoscape_v3.5.1 [41]. Here, within each module, we selected 30 genes whose degree is the largest as Hub genes. In addition, according to the Cufflinks, we calculated the FPKM (fragments per kilobase of transcript sequence per million base pairs sequenced) including 2 biological replicates. Among the differentially spliced genes, we performed an independent-sample t-test based on the FPKM values of the inoculated and mock samples using the SPSS Statistics 22 [42] to identify the differentially expressed genes. Differences between independent samples with a *p*-value <0.05 were considered to be significant. We used the average FPKM values of two biological replicates to draw the isoform expression heatmap of the above differentially expressed genes using R version 3.4.4 program.

### 2.4. Functional Enrichment and Clustering

To identify enriched functional terms for the modules identified by WGCNA, the R package topGO was used to perform Gene Ontology (GO) analysis [43]. The enriched KEGG (Kyoto Encyclopedia of Genes and Genomes) pathways of genes in each module were identified using the Omicshare platform (www.omicshare.com/tools).

## 3. Results

### 3.1. Putative Transcript Assembly

To identify the transcripts and expression patterns of every isoform, we performed the processes described below to obtain the data required for AS analysis. Using the *B. napus* reference genome (http://www.genoscope.cns.fr/brassicanapus/data/), the raw data were quality trimmed, followed by transcript assembly and transcript merging using the RNA-seq data analysis pipeline. We identified 8641, 9614, 8829, 9071, 10,193, 10,035, 11,086, and 5698 new transcripts with the class code “u” in these samples: two replicates of non-inoculated cotyledons, inoculated cotyledons, non-inoculated stems, and inoculated stems (Appendix A). The genes belonging to class “u” are the new transcripts that do not exist at the reference genome according to the Cufflinks. We also obtained the FPKM values of the genes and isoforms. The average FPKM values of the eight materials examined was 15.93, 15.35, 14.63, 14.68, 13.51, 13.86, 12.05, and 13.66, respectively (Appendix A), whereas the average FPKM values of the new transcripts were 6.05, 5.09, 5.39, 5.43, 7.29, 6.98, 6.65, and 7.43, respectively. The gene expression levels were higher in cotyledons than in stems and higher in the mock-inoculated samples than in those inoculated with *L. maculans*.

### 3.2. Analysis of the AS Landscape

With the 101,040 annotated genes in the *B. napus* as the reference genome, we determined the AS patterns and landscape of our samples. As shown in Figure 1A, 14,708 overlapping genes were found between mock-inoculated and inoculated cotyledons, and 8237 and 8873 differentially spliced genes were detected in mock-inoculated and inoculated cotyledons, respectively. The overlapping genes are likely to be under-represented due to the low robustness. In Figure 1B, in total, 63,069 AS events (one AS gene may have more than one AS pattern) were detected in mock-inoculated cotyledons, with intron retention (IR) events being the most common (~31,634; 50%), followed by other (~15,800; 25%), AA (alternative acceptor; ~9505, 15%), AD (alternative donor; ~4998, 8%), and ES (exon skipping; ~1132, 2%) events (Appendix A). In inoculated cotyledons, 61,540 AS events were detected, with IR events being the most common (~31,063; 50%), followed by other (~15,263; 25%), AA (~9269; 15%), AD (~4826, 8%), and ES (~1119, 2%) events.

As shown in Figure 1A, there were 15,353 overlapping genes in stems, including 11,015 differentially spliced genes in mock-inoculated stems and 10,155 in inoculated stems. In Figure 1B, in total, 72,314 AS events were detected in mock-inoculated stems, with IR events being the most common (~36,769; 51%), followed by other (~17,721; 24%), AA (~10,807; 15%), AD (~5652; 8%), and ES (~1365; 2%) events. Finally, 61,934 AS events were detected in inoculated stems, with IR events being the most common (~32,045; 52%), followed by other (~14,641; 23%), AA (~9346; 15%), AD (~4747; 8%), and ES (~1155; 2%) events. These results indicate that the AS events in stems were almost the same as those in cotyledons before and after inoculation and that IR is the most common AS pattern.

We detected 1892 overlapping AS genes in the inoculated materials and identified their homologous genes in *A. thaliana* (Appendix A) [44]. There are 1817 genes exhibiting homology in *A. thaliana*. Again, IR was the most common AS pattern for these genes, followed by other, AA, AD, and ES events. Therefore, AS patterns are highly conserved among genes from different samples.

### 3.3. WGCNA of Overlapping Differentially Spliced Genes

We used WGCNA to further explore the functions of the 1892 overlapping genes, as such an analysis could be used to uncover candidate genes that function in the pathogen response. This technique uses the topological overlap measure (TOM) to cluster similarly expressed genes into discrete modules based on pairwise correlations between genes [45]. Using a soft threshold of 9 (Figure 2), we detected eight modules of genes with highly similar expression patterns. Using a weight value of >0.15, we performed network analysis with Cytoscape to display the relationships of each gene in a single module (Appendix A). Figure 3A–H shows the red, black, blue, brown, yellow, turquoise, green, and pink modules, containing 220, 196, 263, 260, 257, 274, 236, and 181 genes, respectively. The most and least highly connected genes are shown in purple and yellow, respectively.

To explore the functions of the genes in the eight modules, we performed GO and KEGG analysis of the inoculated materials (Appendix A) to identify genes related to metabolism. The significant KEGG pathways (*p* < 0.05) and major GO terms (*p* < 0.01) of genes in the blue module include homologous recombination, citrate cycle (TCA cycle), fructose metabolism, cellular response to gamma and ionizing radiation, leaf formation, adventitious root development, and organic acid biosynthetic process. The pathways/terms of genes in the brown module include cyanoamino acid metabolism, nitrogen compound transport, intracellular protein transport, and cellular protein localization. For the green module, these pathways/terms include RNA degradation and lipoic acid metabolism, protein autoubiquitination, telomeric loop formation, modulation by immune response organism, and modulation by symbiont of host immune response (Figure 4 and Table 1). For the pink module, these pathways/terms include insulin resistance, amino sugar and nucleotide sugar metabolism, regulation of auxin polar transport, glycerolipid metabolic process, phosphatidylinositol dephosphorylation, and response to osmotic stress. Therefore, the genes in these four modules mainly function in response to stress, whereas the functionally enriched categories of genes in the other modules are related to carbon metabolism, photosynthesis, and plant hormones.

### 3.4. The Hub Genes and TFs in the Modules Identified by WGCNA

Using the highest degree value, 120 Hub genes were identified for the eight modules, encoding proteins such as protein kinase, chitinase, reductase, and oxidase (Appendix A). The Hub genes of each module most highly correlated with others in the network represent key factors that function in defense against pathogen attack. The overlapping Hub genes among the four defense-related modules include genes in the DUF family and genes in the protein kinase superfamily. The DUF family includes *BnaCnng76620D*, *BnaA09g51150D*, *BnaAnng22520D,* and *BnaC09g38570D*. These four genes, which have not previously been reported in *B. napus*, might play regulatory roles in response to pathogens.

Since *TFs* are crucial regulators of transcription, we focused on differentially spliced *TF* genes. We identified 180 *TF* genes among the 1892 differentially spliced genes, including 12 *AP2/ERF-ERF*, 10 *bHLH*, 9 *C2H2*, 9 *WRKY*, and 8 *GARP-G2-like*. In addition, among the 120 Hub genes, there are nine genes encoding nine *TFs*. Most *TF* genes are highly connected to other genes (indicated in purple), suggesting that *TFs* are high-level regulators in the networks. The Hub *TF* genes belong to 9 families are including *NAC* (*BnaA06g34140D*), *WRKY* (*BnaC07g09430D*), *TRAF* (*BnaC07g02890D*), *AP2/ERF-ERF* (*BnaC03g14960D*), *C2H2* (*BnaA09g18800D*), *C2C2-GATA* (*BnaC01g44080D*), *HMG* (*BnaA01g01900D*), *bHLH* (*BnaA09g00910D*), and *C2C2-CO-like* (*BnaC05g50730D*) *TFs*.

### 3.5. Identification of Differentially Expressed Genes in Mock-Infected and Infected Samples among Differentially Spliced Genes

The regulatory relationship of gene expression and AS is unknown. Current research indicates that the overlap between differentially expressed genes and differentially alternatively spliced genes is small [46], which makes these overlapping genes particularly worthy of further research. Among the 1892 differentially alternatively spliced genes, we identified significant differentially expressed genes between mock-infected and infected materials. There were 52 and 117 such genes in cotyledons and stems, respectively. To explore their role in defending the pathogen, we constructed a heatmap of the isoforms (134 in cotyledons and 303 in stems) of these differentially expressed genes based on expression patterns (Figure 5A,B). As shown in the heatmap, the different isoforms of a single gene almost always had the same expression pattern; however, there was always a predominant isoform.

We identified four overlapping differentially expressed genes (*BnaA03g18030D*, *BnaA03g54830D*, *BnaCnng49050D,* and *BnaCnng52760D*) in mock-infected versus infected materials in both cotyledons and stems, suggesting that these four genes play important roles in the pathogen response. Based on the results of WGCNA, these four genes belong to the blue, green, black, and green modules. The isoform 5686.1 of *BnaA03g18030D* (*C2C2-Dof*), expressed at a significantly higher level than isoform 5686.2, has the same expression pattern in cotyledons and stems. Among the isoforms of *BnaCnng49050D* (*HMG*), 65,273.1 and 65,273.2 were expressed at high levels in stems and at low levels in cotyledons, suggesting that these isoforms function differently in different tissues. In addition, the isoforms of *BnaA03g54830D* and *BnaCnng52760D* share similar expression patterns. There were two *TF* genes among the four overlapping differentially expressed genes: *BnaA03g54830D* (*C2C2-Dof*) and *BnaCnng49050D* (*HMG*).

The differentially expressed genes in cotyledons included the TFs: C2C2-Dof (BnaA03g54830D), bZIP (BnaA06g29500D), AP2/ERF-ERF (BnaC01g36330D), MYB-related (BnaC03g68390D), TRAF (BnaC05g17140D), TAZ (BnaC06g39410D), GARP-G2-like (BnaC08g07790D), GRAS (BnaC09g52270D), and HMG (BnaCnng49050D). Among the differentially expressed genes in stems, 11 TF genes were identified, including C2C2-Dof (BnaA03g54830D), bHLH (BnaA05g01070D), C2H2 (BnaA05g12370D), SNF2 (BnaA05g20990D), B3-ARF (BnaA08g31250D), GARP-G2-like (BnaAnng01860D), BES1 (BnaAnng18240D), HSF (BnaC03g73070D), AP2/ERF-ERF (BnaC09g49920D), HMG (BnaCnng49050D), and zf-HD (BnaCnng66730D). These TFs might represent crucial regulators of the defense response to pathogens via AS and differential expression.

## 4. Discussion

The defense response of *B. napus* to *L. maculans* can be induced by pathogen elicitors, which are involved in processes such as plant cell wall degradation [47] and toxin biosynthesis [48]. Based on many studies of the defense response of *B. napus* to *L. maculans*, at least 18 major R genes have been identified, and adult plant resistance mediated by the quantitative effects of R genes has been described [49]. Here, we explored new transcripts and changes in the isoform-level AS landscape of *B. napus* infected with *L. maculans.*

Among the 1892 differentially spliced genes identified in this study, we identified their homologous genes in *A. thaliana* to gain their AS events in Riken database (http://rarge.gsc.riken.jp/a_splicing/index.pl) [50]. There are 133 *A. thaliana* genes producing AS (Appendix A), which can validate our study to some extent.

### 4.1. Enriched Pathways and Hub Genes Identified by WGCNA

WGCNA is a method used to classify genes with similar expression patterns. In this study, based on the 1892 differentially spliced genes between mock-inoculated and inoculated materials, we performed WGCNA to identify eight modules of genes and 30 Hub genes that connect all the genes together. In addition, we performed KEGG and GO analyses to predict the functions of genes in each module. In the four modules with genes related to defense to stress, the clustered pathways mainly included fructose metabolism, amino sugar metabolic process, citrate cycle, nitrogen compound transport, glycerolipid metabolic process, and others, which corresponds with previously reported results. The Hub genes identified in this study might play important roles in plant defense, such as genes in the categories DUF, protein kinase, chitinase, reductase, oxidase, and metal transport protein.

Chitin, a component of the fungal cell wall, is a pathogen elicitor that induces *B. napus* to produce nitric oxide and hydrogen peroxide in epidermal cells [51]. The identification of a Hub gene in the chitinase category suggests that plants might degrade pathogen elicitors as a defense response. Furthermore, conserved metabolic pathways such as the glyoxylate cycle, amino acid biosynthesis, and glycerolipid metabolism are essential for pathogenic processes [52]. Lipids play important roles in primary metabolism and represent the main storage form carbohydrates in fungal spores [53]. In addition, *L. maculans* produces a wide range of cell-wall-degrading enzymes [54]. In this study, we identified Hub genes related to the metabolism of fructose, a basic component of the cell wall, suggesting that the invasion of *L. maculans* might lead to the degradation of the cell wall, which may stimulate plants to synthesize new cell wall or alter the cell wall of healthy plant tissues to prevent further pathogen penetration.

The overlapping Hub genes include DUF family members BnaCnng76620D, BnaA09g51150D, BnaAnng22520D, and BnaC09g38570D, whose homologous genes in A. thaliana are AT3G43250 (AtDUF572), AT1G02816 (AtDUF538), AT5G01750 (AtDUF567), and AT2G20625 (AtDUF626), respectively. These genes are mainly involved in abiotic and biotic stress responses. For example, AtDUF572 and AtDUF538 are drought-inducible genes [55]. AtDUF572 is regulated by AvrE or HopM1 during infection with the hemibiotrophic pathogen Pseudomonas syringae pv. tomato DC3000 (Pst DC3000) [56]. AtDUF567 is targeted by the BZR1 TF, and its expression is responsive to brassinosteroids [57]. AtDUF626 is exclusively upregulated by salt stress in roots [58].

### 4.2. The Roles of TFs in the Defense Response to Pathogens among the AS Genes

TFs play crucial roles in plant growth, development, and responses to stress [59], including biotic and abiotic stress. For example, the TF C2C2-GATA, which contains a highly conserved zinc finger DNA-binding domain, is a key regulator of the response of sweet orange (*Citrus sinensis*) to *Xanthomonas campestris* pv. *Vesicatoria* infection [60]. TFs not only initiate or repress gene transcription, they also undergo constitutive and inducible AS. For example, the different isoforms of OsWRKY62 and OsWRKY76 play different roles and perform AS-mediated feedback regulation in the defense response to the blast fungus and the leaf blight bacterium [26]. In this study, we focused on *TF* genes among the differentially spliced genes, including differentially expressed *TF* genes and the *Hub TFs* identified by WGCNA.

We identified 9 and 11 differentially expressed *TFs* between mock-infected and infected cotyledons and stems, respectively, with two overlapping *TFs*, *HMG* (*BnaCnng49050D*) and *C2C2-Dof* (*BnaA03g54830D*). The roles of these *TFs* in *B. napus* are largely unknown. Their homologs in *A. thaliana*, *AT1G206931* and *AT5G02460* (*Dof5.1*) affect leaf axial patterning by promoting Revoluta transcription [61].

We identified 9 additional *TF* genes identified as Hub genes by WGCNA, which might function by regulating the lower hierarchical genes to defend the plant against pathogen attack. We focused on the modules related to pathogen defense, especially the Hub *TFs BnaA06g34140D*, *BnaC07g09430D*, *BnaC07g02890D*, *BnaC03g14960D*, and *BnaA09g18800D*. Their homologs in *A. thaliana* are *AT2G02450* (*AtNAC035*), *AT1G29860* (*AtWRKY71*), *AT5G45110* (*AtNPR3*), *AT5G53290* (*AtCRF3*), and *AT2G01940* (*AtSGR5*), respectively. The overexpression of *AtWRKY71* can affect the defense response to *Pseudomonas syringae* [62], and AtNPR3 functions as an activator to stimulate disease resistance in plants [63]. The Hub genes and expression modules identified in this study provide insight into an agronomically important plant-pathogen interaction and could be modulated through AS to improve resistance of *B. napus* to *L. maculans*.

## Figures and Tables

**Figure 1 genes-10-00296-f001:**
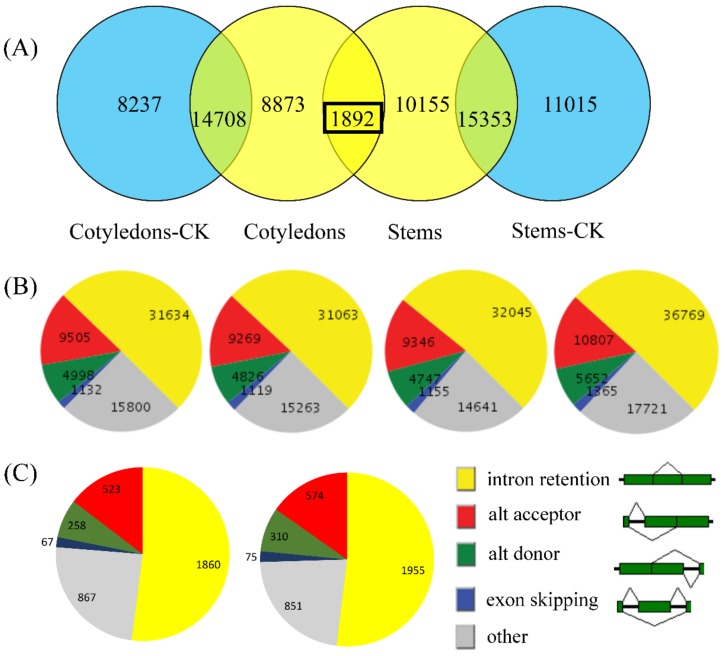
The alternative splicing (AS) landscape of *B. napus* mock-inoculated and inoculated with *L. maculans*. (**A**) Genes exhibiting AS in mock-inoculated cotyledons, inoculated cotyledons, inoculated stems, and mock-inoculated stems. All inoculated samples were inoculated with *L. maculans*. The black rectangle represents the 1892 overlapping genes in inoculated cotyledons and stems. (**B**) The AS patterns of genes in mock-inoculated cotyledons, inoculated cotyledons, inoculated stems, and mock-inoculated stems. (**C**) The AS patterns of the 1892 overlapping genes in inoculated cotyledons and stems.

**Figure 2 genes-10-00296-f002:**
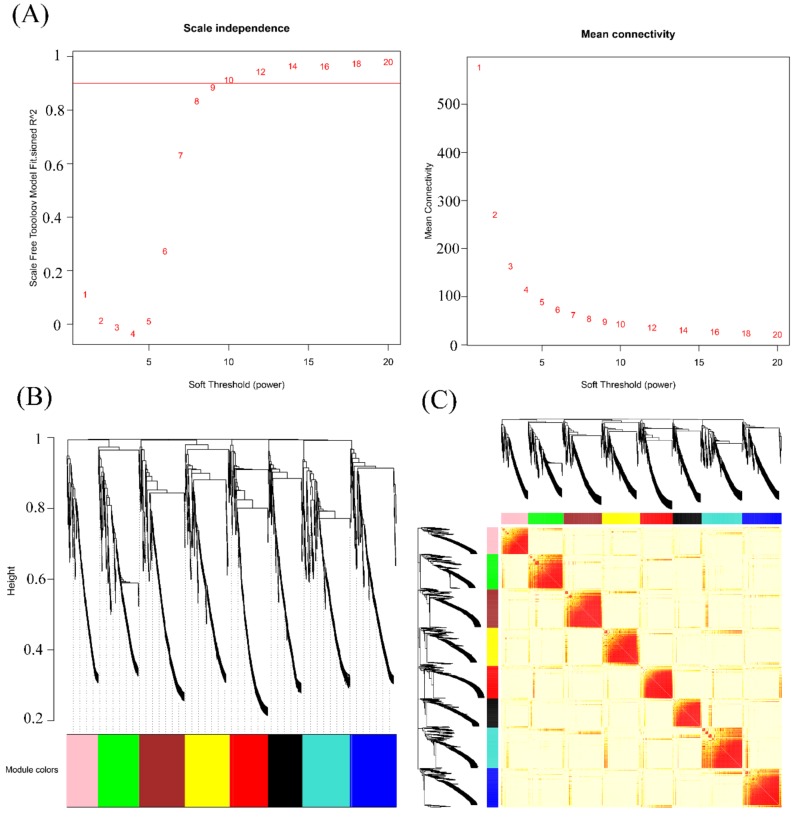
Module identification by WGCNA. (**A**) Selection of the soft threshold with scale independence and mean connectivity. (**B**) Cluster dendrogram of the eight modules. The expression patterns of genes in the gray module are not significantly related, and the genes were not classified into any module. (**C**) Heatmap of the eight modules.

**Figure 3 genes-10-00296-f003:**
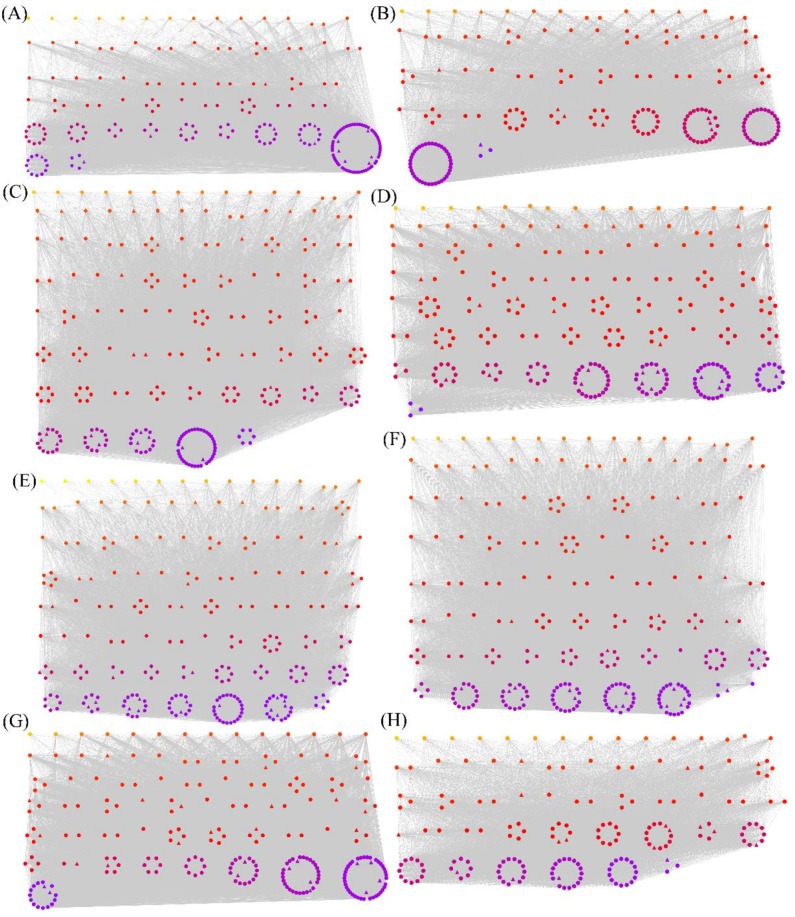
Visualization of the eight co-expression networks by Cytoscape. (**A**) the red, (**B**) black, (**C**) blue, (**D**) brown, (**E**) yellow, (**F**) turquoise, (**G**) green, and (**H**) pink modules, respectively. The most and least highly connected genes are shown in purple and yellow, respectively. The *TFs* are indicated by triangles, whereas the remaining genes are indicated by circles.

**Figure 4 genes-10-00296-f004:**
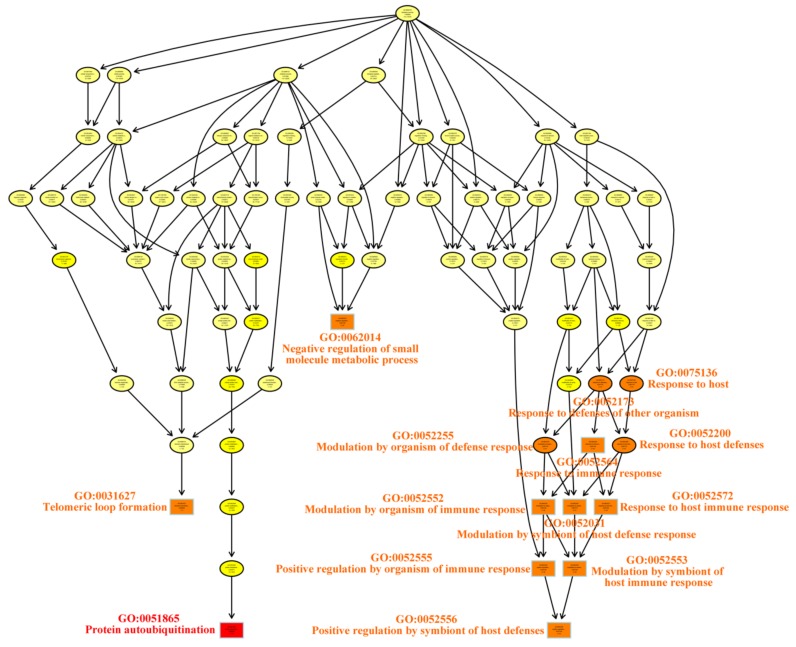
The enriched Gene Ontology (GO) biological process categories of genes in the green module.

**Figure 5 genes-10-00296-f005:**
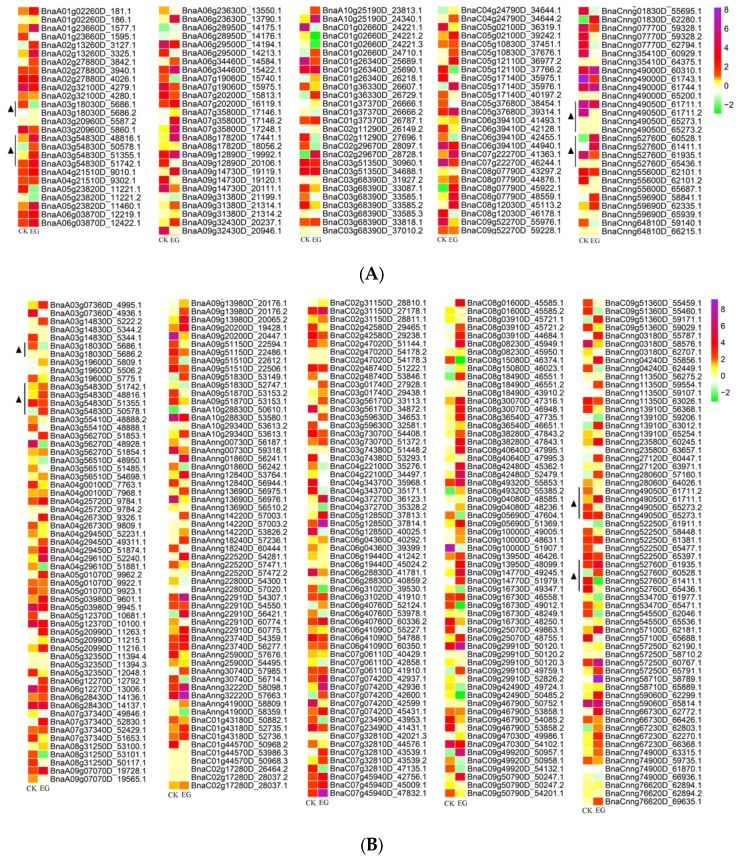
Heatmap of the isoform expression levels of the differentially expressed genes of mock-infected and infected materials among the 1892 differentially spliced genes. (**A**) Heatmap of the differentially expressed genes of mock-infected and infected cotyledons. (**B**) Heatmap of the differentially expressed genes of mock-infected and infected stems. CK and EG represent mock-infected and infected materials, respectively.

**Table 1 genes-10-00296-t001:** The 15 enriched Gene Ontology (GO) terms of genes in the green module according to classic Fisher to order.

Term	Annotation	Significant	Expected	Classic Fisher	
GO:0051865	Protein Autoubiquitination	49	3	0.14	0.00036
GO:0031627	Telomeric Loop Formation	2	1	0.01	0.00560
GO:0052552	Modulation by Organism of Immune Response	45	2	0.13	0.00715
GO:0052553	Modulation by Symbiont of Host Immune Response	45	2	0.13	0.00715
GO:0052555	Positive Regulation by Organism of Immune Response	45	2	0.13	0.00715
GO:0052556	Positive Regulation by Symbiont of Host	45	2	0.13	0.00715
GO:0052564	Response to Immune Response	45	2	0.13	0.00715
GO:0052572	Response to Host Immune Response	45	2	0.13	0.00715
GO:0062014	Negative Regulation of Small Molecule Members	45	2	0.13	0.00715
GO:0052031	Modulation by Symbiont of Host Defense Response	47	2	0.13	0.00777
GO:0052173	Response to Defenses of Other Organism Immune Response	47	2	0.13	0.00777
GO:0052200	Response to Host Defenses	47	2	0.13	0.00777
GO:0052255	Modulation by Organism of Defense Response	47	2	0.13	0.00777
GO:0075136	Response to Host	47	2	0.13	0.00777
GO:0018107	Peptidyl-threonine Phosphorylation	3	1	0.01	0.00838

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
