# Peer review of "The Alternative Splicing Landscape of Brassica napus Infected with Leptosphaeria maculans"

_genes, 2019, doi:10.3390/genes10040296_

Round 1
Reviewer 1 Report
The manuscript “The Alternative Splicing Landscape of Brassica napus Infected with
Leptosphaeria maculans” attempts to discuss alternative splicing mediated posttranscriptional
regulation in B. napus infected with the fungal pathogen Leptosphaeria maculans. I found introduction and discussion sections thoroughly explained. However, I have found “Method” and “Result” sections very shallow and incomplete. Manuscript is full with such typos that have major effect on the interpretation of the sentences/analysis. The text is cluttered at many places. So, there is less continuity in the flow as well as less clarity in the text for readers. More specific and major revisions include:
1. “posttranscriptional” should be replaced with post-transcriptional throughout the manuscript.
2. In line 22-25, rephrase the sentence to highlight specific findings of the co-expression
analysis.
3. In line 28, “exploited” sounds more appropriate in place of manipulated.
4. In line 75-76, it would be highly informative to include some statistics of economic losses due to L. maculans.
5. In line 97, does author mean SRA toolkit? If so, please correct the typo and the reference [29].
6. In subsection 2.2 and other sections, author must mention the version of the all the software used for the analysis.
7. In subsection 2.2 and other sections, author must mention parameters (unless default) used during the analysis steps. This is highly important to reproduce the outcomes of the analysis in future.
8. In line 107 “After filtering”, it is not clear what is filtered? Also what does author mean by weighted values? Please elaborate and explicitly mention in the text the steps to identify hub genes.
9. Ambiguous and long sentence in line 104-107. Please rephrase.
10. In line 111, repeats should be replaced with replicates. Also mention which significant
test of expression differences was performed and why?
11. In line 113, P-value > 0.05 is not significant. Does author mean P-value < 0.05? Please
justify.
12. In sub-section 2.5, two genes are very few to validate the accuracy of AS events
experimentally. Any reason to validate such a low number? Please include more genes
(atleast 20-30) for experimental validation to access the accuracy and strengthen the
current findings.
13. In line 138, what does the author mean by code “u”? Please mention explicitly in the
text.
14. In line 147, did author annotate the genes in B. napus in addition to the assembly? If
yes, the steps and parameters involved for gene annotation is not clearly
mentioned/missing in the method section. Also number of annotated genes is quite
high (101,040). Could author compare the number of genes with other related crops
and comment on the observation?
15. In Fig.1, what does the black rectangle signify, please include in the legend.
16. In line 190, is it the heatmap for pairwise correlation? Else, what values are used to
plot heatmap? Please specify in the text.
17. In figure 4, text is not visible in the circle. Either remove the text or mention it outside
circle with big font size.
18. In line 227, mention how many hub genes were TFs.
19. Did author arbitrarily choose the number of modules obtained in WGCNA analysis?
Please justify.
20. In the supplementary file, there’s inconsistency in the header names at several places.
For e.g., coverage is mentioned as “cov” or “coverage”. Please maintain consistency
in the header for all the files.
Author Response
Dear Reviewers,
Thank you for your kind suggestions and comments. We sincerely appreciate your valuable comments, which not only helped us improve our manuscript, but also provide some good ideas for future research. We have studied your comments carefully and have made the required corrections. We hope that the revised version of our manuscript will meet with your approval. The main corrections and responses to your comments are listed below.
Best regards,
Jiana Li
List of responses:
The manuscript “The Alternative Splicing Landscape of Brassica napus Infected with
Leptosphaeria maculans” attempts to discuss alternative splicing mediated posttranscriptional regulation in B. napus infected with the fungal pathogen Leptosphaeria maculans. I found introduction and discussion sections thoroughly explained. However, I have found “Method” and “Result” sections very shallow and incomplete. Manuscript is full with such typos that have major effect on the interpretation of the sentences/analysis. The text is cluttered at many places. So, there is less continuity in the flow as well as less clarity in the text for readers. More specific and major revisions include:
1. “posttranscriptional” should be replaced with post-transcriptional throughout the manuscript.
Response: I have modified this term as you suggested.
2. In line 22-25, rephrase the sentence to highlight specific findings of the co-expression analysis.
Response: Thank you for your suggestions. We rephrased the sentence in lines 22–27, page 1 “We performed a weighted gene co-expression network analysis (WGCNA) to identify eight co-expression modules and their hub genes, which are the most associated genes with others among each module. Many hub genes belong to the domain of unknown function (DUF) family. Furthermore, some hub genes encode transcription factors, which represent key regulators of each module, including members of the NAC, WRKY, TRAF, AP2/ERF-ERF, C2H2, C2C2-GATA, HMG, bHLH, and C2C2-CO-like families.”
3. In line 28, “exploited” sounds more appropriate in place of manipulated.
Response: Thank you for your suggestions. I have replaced “manipulated” with “exploited”.
4. In line 75-76, it would be highly informative to include some statistics of economic losses due to L. maculans.
Response: Thanks a lot. The following sentences have been added in lines 83–86, page 2 “It was estimated that UK losses of £56M per season reported in 2006 and an average losses of £235M each year (http://www.cropmonitor.co.uk/) for the period 2005-2014 caused by phoma canker (L. maculans and L. biglobosa).”
5. In line 97, does author mean SRA toolkit? If so, please correct the typo and the reference [29].
Response: We apologize for the typo. We modified it into “SRA Toolkit 2.9.0 (https://www.ncbi.nlm.nih.gov/Traces/sra/?view=software)”.
6. In subsection 2.2 and other sections, author must mention the version of
the all the software used for the analysis.
Response: Thanks for your suggestions. We added the version of
the software: “SRA Toolkit 2.9.0”, “Trimmomatic-0.36”, “STAR-2.5.3a”, “Cufflinks-2.2.1”, “ASTALAVISTA-4.0”, “R version 3.4.4”, “Cytoscape_v3.5.1”, and “SPSS Statistics 22”.
7. In subsection 2.2 and other sections, author must mention parameters (unless default) used during the analysis steps. This is highly important to reproduce the outcomes of the analysis in future.
Response: Thank you for the good suggestion. We added the modified parameters as you suggested in subsection 2.2: “Trimmomatic-0.36 (modified parameters: ILLUMINACLIP:TruSeq3-PE.fa:2:30:10, LEADING:3, TRAILING:3, and SLIDINGWINDOW:4:15)”, “STAR-2.5.3a (with sjdbOverhang = 150 and limitBAMsortRAM = 6000000000)”.
8. In line 107 “After filtering”, it is not clear what is filtered? Also what does author mean by weighted values? Please elaborate and explicitly mention in the text the steps to identify hub genes.
Response: Thanks for your comments. It was filtered to obtain gene pairs with weighted values of > 0.15. The weighted values represent the relationships between genes in pairs. We rephrased the text as you suggested. Here are the revised sentences: “From above analysis, we gained the weighted values, which represent the relationships between genes in pairs. The weighted values, which are > 0.15, were used to perform network analysis and gain the genes’ “degree (i.e., the number of genes linked to the gene)” using Cytoscape_v3.5.1. Here, within each module, we selected 30 genes whose degree is the largest as hub genes.”
9. Ambiguous and long sentence in line 104-107. Please rephrase.
Response: Thanks for your comments. We have rephrased it, which was also presented in question 8 “From above analysis, we gained the weighted values, which represent the relationships between genes in pairs. The weighted values, which are > 0.15, were used to perform network analysis and gain the genes’ “degree (i.e., the number of genes linked to the gene)” using Cytoscape_v3.5.1. Here, within each module, we selected 30 genes whose degree is the largest as hub genes.”
10. In line 111, repeats should be replaced with replicates. Also mention which significant test of expression differences was performed and why?
Response: Thank you very much. We replaced “repeat” with “replicate” throughout the manuscript. We also stated that we used an independent-sample t test to identify the differentially expressed genes of mock-infected and infected samples among the 1892 differentially spliced genes.
11. In line 113, P-value > 0.05 is not significant. Does author mean P-value < 0.05? Please justify.
Response: We apologize for this typo. We have corrected it. Thank you for spotting that.
12. In sub-section 2.5, two genes are very few to validate the accuracy of AS events experimentally. Any reason to validate such a low number? Please include more genes (atleast 20-30) for experimental validation to access the accuracy and strengthen the current findings.
Response: We really appreciate your advice. We paid much attention to the experimental validation. However, here are some problems that led to such a low number. First, the pathogen Leptosphaeria maculans is not allowed to be introduced to our country, so we do not have the materials same to the samples which were used to perform RNA-seq analyses, since the data was downloaded from GEO database. The differences of materials made it hard to validate the results. Second, without the same materials, it is impossible to obtain the sequences of different isoforms in our own materials. Therefore, it is hard to design the proper gene primers to amplify alternative splicing isoforms. We apologize for the above problems. After discussing these issues with the co-authors, we decided to delete this section (section 2.5 and section 3.6) from the manuscript. Since the innovation of this manuscript is to provide bioinformatic methods to identify the candidate genes responding the pathogen from two aspects: AS and gene differential expression, we hope that the limitation of the experimental validation can be understood. Thanks for your suggestion again.
13. In line 138, what does the author mean by code “u”? Please mention explicitly in the text.
Response: This refers to new transcripts that do not exist in the reference genome according to Cufflinks software. We added this definition as you suggested.
14. In line 147, did author annotate the genes in B. napus in addition to the assembly? If yes, the steps and parameters involved for gene annotation is not clearly mentioned/missing in the method section. Also number of annotated genes is quite high (101,040). Could author compare the number of genes with other related crops and comment on the observation?
Response: We apologize for the confusing sentences and have corrected them. We only finished the assembly, and not the annotation which was published in Science in 2014, of B. napus. Oilseed rape is an allopolyploid species that arose through spontaneous hybridizations between B. rapa and B. oleracea. Together with more ancient polyploidizations, this conferred an aggregate 72× genome multiplication; therefore, B. napus has more annotated genes than many other plants. Since the observation had been largely known, which is not first presented by us, we think it is unnecessary to compare it with other crops. However, because of your helpful suggestion, we think it is necessary to describe the evolution process of B. napus in the introduction in line 80–81. Thanks a lot.
15. In Fig.1, what does the black rectangle signify, please include in the legend.
Response: We defined the black rectangle as you suggested. Thanks!
16. In line 190, is it the heatmap for pairwise correlation? Else, what values are used to plot heatmap? Please specify in the text.
Response: We apologize for the confusion. The heatmap is based on the isoform expression (FPKM values) of the differentially expressed genes between mock and infected samples. We added this information in section 2.3. Here are the modifications: In addition, according to the Cufflinks, we calculated the FPKM (fragments per kilobase of transcript sequence per million base pairs sequenced) including 2 biological replicates. Among the differentially spliced genes, we performed an independent-sample t test based on the FPKM values of the inoculated and mock samples using the SPSS Statistics 22 to identify the differentially expressed genes. Differences between independent samples with a P-value < 0.05 were considered to be significant. We used the average FPKM values of two biological replicates to draw the isoform expression heatmap of the above differentially expressed genes using R version 3.4.4.
17. In figure 4, text is not visible in the circle. Either remove the text or mention it outside circle with big font size.
Response: Thanks for your suggestion. We mentioned it outside circle with big font size in the figure as you suggested. In addition, there is the details of the enriched GO terms in Table 1 and supplementary file (Table S6).
18. In line 227, mention how many hub genes were TFs.
Response: We added it as you suggested in line 244 “In addition, we identified nine TF genes among the 120 hub genes.” Thanks!
19. Did author arbitrarily choose the number of modules obtained in WGCNA analysis? Please justify.
Response: I am not sure if I understand your question that you wanted to know. We never changed the number of modules (eight modules) which were produced by WGCNA and cannot be determined by ourselves. As for the classification of modules (four defense-related modules and four modules left), it based on the GO and KEGG analyses, for example, the GO terms of module “green” are highly related to plant defense, and the module “blue”, “brown” and “pink” is also related to immunity. This identified process is in lines 191-205. Thanks for your comment.
20. In the supplementary file, there’s inconsistency in the header names at several places. For e.g., coverage is mentioned as “cov” or “coverage”. Please maintain consistency in the header for all the files.
Response: Thanks for your patience. We modified the “coverage”, “length”, “gene_id”, “cuff_id”, the name of excel sheet, the font size and so on.
In the end, we are extremely grateful for your helpful suggestions. Thanks!
Reviewer 2 Report
The work presented here is of interest to the field and basic idea behind the analysis is sound. However, in my opinion the authors lack the statistical power to make their conclusions. This analysis requires more biological replicates for the differential splicing/expression analysis and the WGCNA R package they used explicitly states that at least 15 samples should be included, while the authors have only 8. Specific comments are below:
2: 69
Iron deficiency citation seems superfluous. If a specific example is desired for this paragraph, using a biotic example would be better.
3:89
Two biological replicates is not enough to generate trustworthy gene expression tests, let alone splicing tests which require more replicates and preferably deeply sequenced paired-end libraries.
3: 103
WGCNA relies heavily on using a relatively large number of samples. The authors of the WGCNA R package recommend a minimum of 15 samples (https://horvath.genetics.ucla.edu/html/CoexpressionNetwork/Rpackages/WGCNA/faq.html). The authors of this paper utilized only 8, and each sample contained only 2 biological replicates. There is unlikely to be enough statistical significance to draw firm conclusions without additional samples.
3: 122
Two genes is not enough to confirm the splicing analysis given that only 2 replicates were used for the RNA-seq data.
4:147
How exactly were overlaps determined? If a gene barely passed the p value cutoff in one sample and barely failed it in another would this be listed as non-overlapping? This is a common issue in this type of analysis which I won't expect authors to necessarily resolve but it may be useful to mention that the overlap is likely under-represented.
9: 242
This meaning of this sentence is unclear to me. It seems to imply gene expression is controlled by AS?
11: 307
The plant's goal here could also be cell wall alteration of healthy tissues to prevent further pathogen penetration
Author Response
Dear Reviewers:
Thank you for your kind suggestions and comments. We sincerely appreciate your valuable comments, which not only helped us improve our manuscript, but also provide some good ideas for future research. We have studied your comments carefully and have made the required corrections. We hope that the revised version of our manuscript will meet with your approval. The main corrections and responses to your comments are listed below.
Best regards,
Jiana Li
The work presented here is of interest to the field and basic idea behind the analysis is sound. However, in my opinion the authors lack the statistical power to make their conclusions. This analysis requires more biological replicates for the differential splicing/expression analysis and the WGCNA R package they used explicitly states that at least 15 samples should be included, while the authors have only 8. Specific comments are below:
2: 69
Iron deficiency citation seems superfluous. If a specific example is desired for this paragraph, using a biotic example would be better.
Response: Thanks for your suggestions! We have replaced this session with a biotic example as you suggested in lined 75–76, page2 “For example, it was reported that constitutive and inducible alternative splicing of OsWRKY62 and OsWRKY76 may defend the blast fungus Magnaporthe oryzae and the leaf blight bacterium Xanthomonas oryzae pv oryzae.”
3:89
Two biological replicates is not enough to generate trustworthy gene expression tests, let alone splicing tests which require more replicates and preferably deeply sequenced paired-end libraries.
Response: We really appreciate your comments! I strongly agree that more biological replicates lead to more trustworthy results. However, the two biological replicates here can also produce credible results. As for gene expression tests, there is an article (The Transcriptional Landscape of the Yeast Genome Defined by RNA Sequencing) published in SCIENCE, which only used two biological replicates to perform RNA-seq. In addition, only two biological replicates were used to generate gene expression tests in the article (Genome-Wide Association and Transcriptome Analyses Reveal Candidate Genes Underlying Yield-determining Traits in Brassica napus) which was published in Frontiers in Plant Science in 2017 and the article (A combination of genome-wide association and transcriptome analysis reveals candidate genes controlling harvest index-related traits in Brassica napus) published in Scientific Reports in 2016 and so on. As forsplicing tests, there is an article (RNA-Seq of Arabidopsis Pollen Uncovers Novel Transcription and Alternative Splicing) published in Plant Physiology in 2013, whichonly used two biological replicates perform alternative splicing analysis. In terms of this manuscripts, the RNA-seq data that we analyzed here was performed by Genoscope (Evry, France) using Illumina HiSeq2500 to generate 100-bp paired-read sequencing, which is preferably deeply sequenced data. Thanks for your suggestion again!
3: 103
WGCNA relies heavily on using a relatively large number of samples. The authors of the WGCNA R package recommend a minimum of 15 samples (https://horvath.genetics.ucla.edu/html/CoexpressionNetwork/Rpackages/WGCNA/faq.html). The authors of this paper utilized only 8, and each sample contained only 2 biological replicates. There is unlikely to be enough statistical significance to draw firm conclusions without additional samples.
Response: We really appreciate your comments! A sample number of at least 15 is recommended; however, this is a recommendation and not a requirement. For example, there is an article (Comparative transcriptomics reveals patterns of selection in domesticated and wild tomato) published in PNAS that only used 10 samples without biological replicates to perform WGCNA. You can find the sample information in page 24 of the Supplementary Information Appendix of the above paper. In terms of our own results, the WGCNA results showed that our samples are enough good to perform this analysis. As shown in Fig. 2a), the scale independence and mean connectivity is standard, which indicates this network can be regarded as scale-free network. In addition, the scale-free topology fit index reached values of above 0.9 for power (soft threshold = 9, which is less than 15), which indicates that the network is trustworthy to some extent.
3: 122
Two genes is not enough to confirm the splicing analysis given that only 2 replicates were used for the RNA-seq data.
Response: We really appreciate your advice. We paid much attention to the experimental validation. However, here are some problems that led to such a low number. First, the pathogen Leptosphaeria maculans is not allowed to be introduced to our country, so we do not have the materials same to the samples which were used to perform RNA-seq analyses, since the data was downloaded from GEO database. The differences of materials made it hard to validate the results. Second, without the same materials, it is impossible to obtain the sequences of different isoforms in our own materials. Therefore, it is hard to design the proper gene primers to amplify alternative splicing isoforms. We apologize for the above problems. After discussing these issues with the co-authors, we decided to delete this section (section 2.5 and section 3.6) from the manuscript. Since the innovation of this manuscript is to provide bioinformatic methods to identify the candidate genes responding the pathogen from two aspects: AS and gene differential expression, we hope that the limitation of the experimental validation can be understood. Thanks for your suggestion again.
4:147
How exactly were overlaps determined? If a gene barely passed the p value cutoff in one sample and barely failed it in another would this be listed as non-overlapping? This is a common issue in this type of analysis which I won't expect authors to necessarily resolve but it may be useful to mention that the overlap is likely under-represented.
Response: This is an excellent suggestion. The situation that you presented is likely to happen. We are thankful for your understanding. We mentioned that the overlap is likely to be under-represented as you suggested in the manuscript. Thanks for your comments again!
9: 242
This meaning of this sentence is unclear to me. It seems to imply gene expression is controlled by AS?
Response: We apologize for the confusing sentences. We did not mean to imply that gene expression is controlled by AS. It is your kind comment that made us realize the incorrect conclusion! We rephrased these sentences in section 3.5 as follows: (1) lines 252–254: The regulatory relationship of gene expression and AS is unknown. Current research indicates that the overlap between differentially expressed genes and differentially alternatively spliced genes is small, which makes these overlapping genes particularly worthy of further research. (2) Lines 268–270: The isoform 5686.1 of BnaA03g18030D (C2C2-Dof), expressed at a significantly higher level than isoform 5686.2, has the same expression pattern in cotyledons and stems.
11: 307
The plant's goal here could also be cell wall alteration of healthy tissues to prevent further pathogen penetration.
Response: Thanks for your suggestion! We added this goal in lines 325–327, page 11 “The invasion of L. maculans might lead to the degradation of the cell wall, which may stimulate plants to synthesize new cell wall or alter the cell wall of healthy plant tissues to prevent further pathogen penetration.”
In the end, we are extremely grateful for your helpful suggestions. Thanks!
Reviewer 3 Report
-Please the comments on the margins of the manuscript.
-I would highly recommend validation of the more genes preferably couple of genes from each module
- Further, given number of alternatively spliced genes and their isoforms showing differential expression it is essential to show the in gel assay of the alternatively spliced forms from each hub under mock and pathogen infected conditions.
Author Response
Dear Reviewers:
Thank you for your kind suggestions and comments. We sincerely appreciate your valuable comments, which not only helped us improve our manuscript, but also provide some good ideas for future research. We have studied your comments carefully and have made the required corrections. We hope that the revised version of our manuscript will meet with your approval. The main corrections and responses to your comments are listed below.
Best regards,
Jiana Li
-Please the comments on the margins of the manuscript.
Response: We used "Track Changes" function in Microsoft Word to make revisions.
-I would highly recommend validation of the more genes preferably couple of
genes from each module
Response: We really appreciate your advice. We paid much attention to the experimental validation. However, here are some problems that led to such a low number of genes to be validated. First, the pathogen Leptosphaeria maculans is not allowed to be introduced to our country, so we do not have the materials same to the samples which were used to perform RNA-seq analyses, since the data was downloaded from GEO database. The differences of materials made it hard to validate the results. Second, without the same materials, it is impossible to obtain the sequences of different isoforms in our own materials. Therefore, it is hard to design the proper gene primers to amplify alternative splicing isoforms. We apologize for the above problems. After discussing these issues with the co-authors, we decided to delete this section (section 2.5 and section 3.6) from the manuscript. Since the innovation of this manuscript is to provide bioinformatic methods to identify the candidate genes responding the pathogen from two aspects: AS and gene differential expression, we hope that the limitation of the experimental validation can be understood. Thanks for your suggestion again.
- Further, given number of alternatively spliced genes and their isoforms showing differential expression it is essential to show the in gel assay of the alternatively spliced forms from each hub under mock and pathogen infected conditions.
Response: Thanks for your suggestion. It is very useful for our research in the future. However, same to the above response, we are so sorry that we cannot finish the experimental validation because of the limitation of materials. We decided to delete this section from the manuscript as we presented above. We hope that the limitation of the experimental validation can be understood, since we focused on the bioinformatic methods to identify the candidate genes. Thanks for your helpful suggestion.
In the end, we are extremely grateful for your suggestions. Thanks!
Round 2
Reviewer 1 Report
The manuscript “The Alternative Splicing Landscape of Brassica napus Infected with
Leptosphaeria maculans” attempts to discuss alternative splicing mediated posttranscriptional
regulation in B. napus infected with the fungal pathogen Leptosphaeria
maculans. In the revised version of manuscript, majority of the suggested revisions has been
taken care by the authors. There is continuity in the text and clarity in the flow for readers. I
appreciate that the authors have added references wherever suggested as well as corrected
typos. Below listed are the specific revisions:
1. I would recommend authors to concise/improve abstract. My suggestions include:
a. Hub is more about connections. In line 23, I would change “associated genes”
with “connected genes”.
b. I suggest authors to rephrase the sentence from 23-27 to become more
specific to provide number of genes instead of “many, some etc. Also remove
the word, “furthermore”. I would also suggest to remove the sentence
“Many overlapping hub genes belong to the domain of unknown function
(DUF) family” as it doesn’t contain important information to be included in
the abstract.
c. In the line from 27-30 needs rephrasing with more specific information.
2. The last paragraph of introduction needs to be elaborated with the findings of the
current study in brief and mention specifically how could these findings enhance our
understanding of B. napus.
3. From line 175-182, I would suggest authors to make a bar plot with the comparisons
to make it clearer and more informative. So many numbers in between the text
clutter the sentences.
4. In line 227, mention how many hub genes were TFs.
5. In line 147, the number of annotated genes is quite high (101,040). Could author
compare the number of genes with other related crops and comment on the
observation?
6. In the revised version, author removed the functional validation (provided in the first
version). Please comment why can’t author perform functional validation? In any case,
I would recommend authors to incorporate existing literature/evidences to support
the current findings.
Author Response
Dear Reviewers,
Thank you for your kind suggestions and comments. We sincerely appreciate your valuable comments, which not only helped us improve our manuscript, but also provide some good ideas for future research. We have studied your comments carefully and have made the required corrections. We hope that the revised version of our manuscript will meet with your approval. The main corrections and responses to your comments are listed below.
Best regards,
Jiana Li
List of responses:
The manuscript “The Alternative Splicing Landscape of Brassica napus Infected with Leptosphaeria maculans” attempts to discuss alternative splicing mediated posttranscriptional regulation in B. napus infected with the fungal pathogen Leptosphaeria maculans. In the revised version of manuscript, majority of the suggested revisions has been taken care by the authors. There is continuity in the text and clarity in the flow for readers. I appreciate that the authors have added references wherever suggested as well as corrected typos. Below listed are the specific revisions:
1. I would recommend authors to concise/improve abstract. My suggestions include:
a. Hub is more about connections. In line 23, I would change “associated genes” with “connected genes”.
Response: Thank you very much. We have changed it as you suggested.
b. I suggest authors to rephrase the sentence from 23-27 to become more specific to provide number of genes instead of “many, some etc. Also remove the word, “furthermore”. I would also suggest to remove the sentence “Many overlapping hub genes belong to the domain of unknown function (DUF) family” as it doesn’t contain important information to be included in the abstract.
Response: Thank you a lot. We removed the word and the sentence in line 23-25 as you suggested. We rephrased the sentence in line 25 “There are nine hub genes encoding nine transcription factors.”
c. In the line from 27-30 needs rephrasing with more specific information.
Response: Thanks for your suggestion. We rephrased the sentence in line 28 “52 and 117 alternatively spliced genes in cotyledons and stems were also differentially expressed between mock-infected and infected materials.”
2. The last paragraph of introduction needs to be elaborated with the findings of the current study in brief and mention specifically how could these findings enhance our understanding of B. napus.
Response: Thanks for your comments. We revised it as you suggested in line 94-97 “Current studies have exploited the molecular mechanism, the function and research method of AS in many plants, especially in the model plant A. thaliana which is an ancestral karyotype to B. napus. In this study, we present an isoform-level AS landscape of B. napus infected with L. maculans to extend AS research and shed light on the response of B. napus to this agronomically important pathogen.”
3. From line 175-182, I would suggest authors to make a bar plot with the comparisons to make it clearer and more informative. So many numbers in between the text clutter the sentences.
Response: Thanks for your suggestion. All the numbers were presented in Figure 1a) which is a pie chart. In order to make it easy to find, we mentioned the relevant figures in text in line 171, 174, 181 and 182.
4. In line 227, mention how many hub genes were TFs.
Response: We mentioned it as you suggested in line 252-253 “In addition, among the 120 hub genes, there are nine genes encoding nine TFs.” Thanks!
5. In line 147, the number of annotated genes is quite high (101,040). Could author compare the number of genes with other related crops and comment on the observation?
Response: Since the observation had been largely known, which is not presented by us and not the focus of our study, we think it is unnecessary to comment on the observation, so we only mentioned the number of annotated genes in its ancestors: B. rapa and B. oleracea in line 82-83 “There are 101040, 41174 and 45758 annotated genes of B. napus, B. rapa [27]and B.oleracea [28].” Thanks again.
6. In the revised version, author removed the functional validation (provided in the first version). Please comment why can’t author perform functional validation? In any case, I would recommend authors to incorporate existing literature/evidences to support the current findings.
Response: We really appreciate your suggestion. As we explained in the previous response, the pathogen L. maculans is not allowed to be introduced to our country, so we do not have the materials same to the samples which were used to perform RNA-seq analyses, since the data was downloaded from GEO database. The differences of materials made it hard to validate the results. We apologize for it. Since the innovation of this manuscript is to provide bioinformatic methods to identify the candidate genes responding the pathogen from two aspects: AS and gene differential expression, we hope that the limitation of the experimental validation can be understood.
As for incorporating existing literature to support our findings, we discussed existing evidences in the section 4.1 and 4.2. We added existing evidences to further support our study, which was to search the AS events of the homologous genes in A. thaliana of our AS genes. It can verify our results in another plant to some extent. We added the relevant methods in section 2.2 in line 116-117 “In order to identify the homologous genes in A. thaliana of the differentially spliced genes in B. napus, we performed blast analysis using TAIR database (https://www.arabidopsis.org/index.jsp).” We added the relevant results in section 3.2 in line 190-192 “We detected 1892 overlapping AS genes in the inoculated materials and identified their homologous genes in A. thaliana (Table S4). There are 1817 genes existing homology in A. thaliana.” We also added the relevant discussion in section 4 in line 315-318 “Among the 1892 differentially spliced genes identified in this study, we identified their homologous genes in A. thaliana to gain their AS events in Riken database (http://rarge.gsc.riken.jp/a_splicing/index.pl). There are 133 A. thaliana genes producing AS (Table S4), which can valid our study to some extent.”
Reviewer 2 Report
The authors addressed the majority of my concerns. The lack of additional biological replicates is still a problem to me. The authors cite several papers to defend this but 2 of them are quite old (the Science paper is from 2008) and the vast majority of recent publications in the field utilize a minimum of 3 biological replicates. Still, I understand the authors' limited ability to correct this issue.
Author Response
Dear Reviewers,
Thank you for your kind suggestions and comments. We sincerely appreciate your valuable comments, which not only helped us improve our manuscript, but also provide some good ideas for future research. We have studied your comments carefully and have made the required corrections. We hope that the revised version of our manuscript will meet with your approval. The main corrections and responses to your comments are listed below.
Best regards,
Jiana Li
List of responses:
The authors addressed the majority of my concerns. The lack of additional biological replicates is still a problem to me. The authors cite several papers to defend this but 2 of them are quite old (the Science paper is from 2008) and the vast majority of recent publications in the field utilize a minimum of 3 biological replicates. Still, I understand the authors' limited ability to correct this issue.
Response: Thanks for your understanding. We added existing evidences to further support our study, which was to search the AS events of the homologous genes in A. thaliana of our AS genes. It can verify our results in another plant to some extent. We added the relevant methods and results in line 116-117, in line 190-192 and in line 315-318. Thanks again.